

## Using wavelet transform and dynamic time warping to identify the limitations of the CNN model as an air quality forecasting system

Ebrahim Eslami[1], Yunsoo Choi[1,*], Yannic Lops[1], Alqamah Sayeed[1], Ahmed Khan Salman[1]

[1]Department of Earth and Atmospheric Sciences, University of Houston, Houston, TX 77204, United States

*Corresponding author: ychoi6@uh.edu





**Abstract:**

    As the deep learning algorithm has become a popular data analytic technique, atmospheric scientists should have a balanced perception of its strengths and limitations so that they can provide a powerful analysis of complex data with well-established procedures. Despite the enormous success of the algorithm in numerous applications, certain issues related to its applications in air quality forecasting (AQF) require further analysis and discussion. This study addresses significant limitations of an advanced deep learning algorithm, the convolutional neural network (CNN), in two common applications: (i) a real-time AQF model, and (ii) a post-processing tool in a dynamical AQF model, the Community Multi-scale Air Quality Model (CMAQ). In both cases, the CNN model shows promising accuracy for ozone prediction 24 hours in advance in both the United States and South Korea (with an overall index of agreement exceeding 0.8). For the first case, we use the wavelet transform to determine the reasons behind the poor performance of CNN during the nighttime, cold months, and high ozone episodes. We find that when fine wavelet modes (hourly and daily) are relatively weak or when coarse wavelet modes (weekly) are strong, the CNN model produces less accurate forecasts. For the second case, we use the dynamic time warping (DTW) distance analysis to compare post-processed results with their CMAQ counterparts (as a base model). For CMAQ results that show a consistent DTW distance from the observation, the post-processing approach properly addresses the modeling bias with predicted IOAs exceeding 0.85. When the DTW distance of CMAQ-vs-observation is irregular, the post-processing approach is unlikely to perform satisfactorily. Awareness of the limitations in CNN models will enable scientists to develop more accurate regional or local air quality forecasting systems by identifying the affecting factors in high concentration episodes.

**Keyword:** machine learning, neural networks, atmospheric chemistry, air quality modeling.





## 1. Introduction:

Currently, atmospheric scientists have shown significant interest in applying machine learning (ML) algorithms in their field, specifically for air quality forecasting, remote sensing data retrieval, and hurricane tracking. ML is a technique used for developing data-driven algorithms that learn to mimic human behavior on the basis of a prior example or experience. It is a tool that allows systems to more effectively deal with knowledge-intensive problems in complex domains, which occurs via learning that involves gathering information from a training dataset and using certain logic to purposefully detect a pattern of behavior. The fundamental goal of ML models is to apply the detected patterns to make generalizations beyond the examples in the training set.

Generalizations stemming from ML models provide a scope of improvement in a number of physical applications. Evidence of the growing interest in applying ML is the rapid increase in the number of scientific publications in this area, illustrated in Fig. S1. Inevitably, a consequence of such enthusiasm in the field is the risk of exaggerated expectations, fueled by results focusing on the general performance of ML models compared to that of conventional statistical models. Such examples can be found in studies by Eslami et al. (2019a, 2019b, 2019c), Choi et al. (2019), Sayeed et al. (2020), and Lops et al. (2019). To assume more reasonable expectations, we must first explore the current challenges we face when forecasting ambient air quality and then assess how or even whether ML models can address the challenges to produce more accurate forecasting.

To develop a capable air quality forecasting tool, atmospheric scientists often turn to chemical transport models (CTMs) and statistical models, both of which use meteorological parameters and chemical precursors from previous atmospheric conditions to estimate the following conditions. A brief summary of these models appears in Zhang et al. (2012). Although CTMs, with their dynamical implementation of atmospheric chemistry and physics, have shown promise in forecasting, they are too computationally intensive for operational real-time forecasts. Thus, computationally efficient statistical models such as ML have emerged as alternative approaches. Unlike CTMs, however, these models mainly rely on data from a network of monitoring stations that are sparsely distributed and measure a limited number of meteorology and air quality variables (Eslami et al., 2019a). Given the complexity of the formation/depletion of air pollutants such as ozone, this limitation may be vital in predicting extreme events (Eslami et al., 2019b).

Another challenge in predicting ozone concentration is the "external" relationships among predictors. For instance, as important meteorological parameters, temperature and solar radiation are synoptic factors while the wind field is influenced by regional factors such as geography and urbanization. Such conditions particularly affect ozone variability since locally-produced $NO_2$ emissions under certain meteorological circumstances lead to the formation of ozone that is later transported by the wind and detected by monitoring stations (Pan et al., 2015). Nevertheless, station-specific ML models use such chemical and meteorological variables as a footprint of local conditions.

Although local emissions of ozone precursors are the dominant source of ozone, particularly in urban areas, ozone pollution arising from sources outside of a target region, such as background ozone, inevitably degrade local air quality (Camalier et al., 2007). The lack of measurable environmental variables that indicate the potential long-range transport of air pollutants poses an unprecedented challenge for a ML model to estimate ozone concentrations over downwind communities (Eslami et al., 2019a). Because of the nonlinear spatial relationships between neighboring monitoring stations, ML models as operational real-time forecasting systems produce relative uncertainty.





A number of studies have proposed solutions addressing the above limitations of ML
models. Eslami et al. (2019a) implement a deep convolutional neural network (CNN) (Krizhevsky
et al., 2012) model that uses hourly values of several meteorological and air pollution variables to
predict hourly ozone concentrations 24 hours in advance. Even though the accuracy of the
forecasting system guarantees a reasonable level of accuracy, it fails to address high ozone
episodes owing to the infrequent occurrences of such events, which lead to the undertraining of
the CNN model. In another study, Eslami et al. (2019b) propose a data ensemble approach that
mitigates this issue by regularizing the training dataset toward capturing high ozone episodes.
While the authors remove a significant portion of the underprediction biases of the CNN model,
its predictions of ozone during the nighttime and on rainy days are unreliable. Sayeed et al. (2020)
use historical data covering a longer period within a diverse geographical domain (Texas) to train
a similar CNN model. Their results from stations for which fewer measurements are available,
while more accurate, are prone to uncertainty. Using the outputs of air quality and meteorological
forecast models to map the hourly ozone concentrations at station locations, Choi et al. (2019)
train a similar deep CNN model, a spatially generalized model that bias-corrects ozone forecasts
of the community multi-scale air quality (CMAQ) model for all monitoring stations in the EPA
AirNow network. Even though the model significantly improved CAMQ forecasts, the bias-
correction process and the unbalanced CMAQ modeling outputs are unclear.
that uses wavelet transform and dynamic time warping (DTW) to
This paper discusses the general inability of the machine learning model using wavelet
transform and dynamic time warping (DTW). Wavelet transform is a powerful technique for
analyzing the temporal variation of a time-series (Grinsted et al., 2004). Wavelet analysis uses an
adjustable resolution to translate time-series data and then decomposes the data into a certain
frequency level that cannot be achieved by other conventional methods such as Fourier analysis
(Huang et al., 2010). DTW is a non-linear technique that measures any alignment between two
times-series (i.e., model prediction and observation in this study) by warping them to match their
similarities (Berndt and Clifford, 1994). By introducing two applications of CNN in the real-time
ozone forecasting system, we use these analytical tools to identify the source of the prediction
biases of the CNN model. In this paper, we do not describe the forecasting results in detail but
instead refer the reader to studies by Eslami et al. (2019a, 2019b), Choi et al. (2019), and Sayeed
et al. (2020).

**2.  Materials and Methods**
**2.1. Deep convolutional neural networks:**
The deep CNN model (Krizhevsky et al., 2012) is a common deep learning architecture
that has long been used in numerous applications (Deng and Yu, 2014; Schmidhuber, 2015;
Goodfellow et al., 2016; Litjens et al., 2017; Chen et al., 2018; Kamilaris and Prenafeta-Boldú,
2018; Higham and Higham, 2019). Unlike other methods, the CNN model is capable of analyzing
joint features and attaining greater accuracy on large-scale datasets. Deep CNNs can be trained to
approximate smooth, highly nonlinear functions (LeCun et al., 2015), rendering them appropriate
for analyzing nonlinear processes in the atmosphere. In addition, feature extraction using deep
learning algorithms is more efficient than using other neural network methods, particularly when
multiple hidden layers are structured (Krizhevsky et al., 2012).
A schematic for the deep CNN used in this paper appears in Fig. 3. The figure shows the
input layer of the CNN algorithm, which represents the normalized time series of all input





variables. The normalization process prevents a steep cost function and averts one feature
overbearing others. A filter passes through a set of units located in a small neighborhood in the
previous convolutional layer. With local receptive fields, neurons can extract the elementary
features of inputs that are then combined with those of higher layers. The outputs of such a set of
neurons constitute a feature map (see Fig. 3). At each position, various types of units in different
feature maps compute various types of features. A sequential implementation of this procedure for
each feature map is used for scanning the input data with a single neuron in a local receptive field
and storing the states of this neuron at corresponding locations in the feature map. The constrained
units in a feature map perform the same operation on different instances in a time series, and
several feature maps (with different weight vectors) can comprise one convolutional layer. Thus,
multiple features can be extracted in each instance. Once a feature is detected, its exact "location"
becomes less important as long as its approximate position relative to the other features is
preserved (Krizhevsky et al., 2012; LeCun et al., 2015).
CNN uses a kernel of a given size to capture changes in the temporal variation of the input
data by sweeping through time series. The various sections of the data are represented by feature
maps. An additional layer performs local averaging, called "pooling," and subsampling reduces
the resolution of the feature map and the sensitivity of the output to possible shifts and distortions.
This step could potentially discard important information (e.g., sudden ozone peaks) as explained
in Sabour et al. (2017). Hence, this study uses the convolution layer without pooling. The feature
maps are connected to a fully-connected layer, which helps us to map each feature of multiple
inputs to the hourly ozone output (see Fig. 1).
Compared to fully-connected multilayer perceptrons (MLPs) and recurrent neural
networks (RNN), which have been extensively used as regression models, CNNs are attractive for
several reasons. MLPs and RNNs are not explicitly designed to model variance within an
estimation that results from a complex interaction between several inputs and outputs. While MLPs
of sufficient size could indeed capture invariance, they require large networks with a large training
set. Compared to the CNNs proposed in this study, RNNs are challenging to implement and
computationally expensive (Eslami et al., 2019a; Sayeed et al., 2020; Lops et al., 2019).

**2.2. Wavelet transform:**
Wavelet transformation decomposes a signal into a scale frequency space, allowing the
determination of the relative contributions of each temporal scale present within a signal (Mallet,
1989). Wavelet decompositions are powerful tools for analyzing the variation in signal properties
across different resolutions of geophysical variables (Mallet, 1989; Grinsted et al., 2004; Foufoula-
Georgiou and Kumar, 2014). Using a fully scalable modulated window that shifts along with the
signal, the wavelet transform overcomes the inability of the Fourier transform to represent a signal
in the time and frequency domain at the same time (see Fig. S2 in the supplementary document).
The spectrum is calculated for every position. After repeating the process, each time with a
different window size, the results constitute a collection of time-frequency representations of the
signal, all with different resolutions. The data are separated into multiresolution components, each
of which is studied with a resolution that matches its scale (Aiazzi et al., 2002). While high-
resolution components capture fine-scale features in the signal, low-resolution components capture
the coarse-scale features.
As wavelet analysis represents any arbitrary (nonlinear) function by a linear combination
of a set of wavelets or alternative basis functions, they are highly suitable for use as both an
integration kernel for analysis to extract information about the process and a basis for
representation or characterization of processes (Kaheil et al., 2008). Figure S3 in the
supplementary document shows the hourly ozone time series of a monitoring station in downtown
Seoul, South Korea, with wavelet transform for the year 2017. Here, the wavelet transform exhibits
strong power levels associated with period=24 and period=168 in the middle of the year, indicating
dominant daily (24 hours) and weekly variation (168 hours).
**2.3. Dynamic time warping:**
To assess the similarity between two time series, DTW expands or contracts a given time
series to minimize the difference between the two of them (Berndt and Clifford, 1994). Advantage
it has over Euclidean distance, a conventional distance analysis method, is that it highlights when
a shift (e.g., a time lag) occurs between two time-steps in two time series (see Fig. S4 in the
supplementary document). Euclidean distance takes pairs of data within the time series and
compares them. DTW calculates the smallest distance between all points, matching one time-step
to many counterpart steps on the linked time series (see Fig. S4). Owing to its non-linear mapping
capability, it is widely used in various domains from time-series classification (Jeong et al., 2011)
to bioinformatics (Giorgino, 2009), health signal processing (Tormene et al., 2009), and speech
recognition (Berndt and Clifford, 1994).
One benefit of DTW is that it will classify two time series of the same shape as similar
even if their absolute values differ or if one time series contains large variability. Figure S5
compares the DTW distance between the observation time series and two prediction models for an
ozone monitoring station in Texas. DTW detects the differences between CMAQ estimation and
observation with the highest difference in the middle of 2014.

**3.   Results and Discussion**
**3.1. Case 1: CNN as a real-time ozone forecasting system**
In this case, we used the modeling experience reported in Eslami et al. (2019a). Briefly,
the system employs a deep CNN model that uses an hourly variation of seven meteorological and
two air quality parameters from the day before as inputs to predict hourly ozone concentrations on
the following day for 25 monitoring stations in Seoul, South Korea. Figures S7 and S8 show the
accuracy of the CNN model (using the index of agreement (IOA)) and the time series comparison
of average ozone concentrations between the observation and the CNN prediction, respectively.
While the model maintained a proper level of prediction accuracy, it was prone to two main
limitations: (i) Its performance at various times of the year varied (see Fig. S6); and (ii) nighttime
predictions showed higher relative bias and lower modeling performance than daytime predictions
(see Fig. S7). In general, wavelet transform can explain varying, time-dependent modeling
performance; nevertheless, the significant difference between modeling performance during the
daytime and the nighttime indicates an undertrained CNN model.

**3.1.1. Time-dependent model performance:**
The performance of the CNN model is directly dependent on how well the model
understands the relationship between the inputs (meteorology and ozone precursors) and output
(ozone concentration). While emission sources from volatile organic compounds (VOCs) and NOx
are relatively constant in time, meteorological variables govern the variation of the ozone at
different times throughout the year (Choi, 2014; Pan et al., 2019). Temperature, wind speed, and
relative humidity (RH) are among the most important meteorological parameters affecting ozone
variation.





Figure 2 shows the wavelet power transform of the aforementioned meteorological
variables for 2017. Since we used an hourly time series to calculate the wavelet powers, both the
index and the period are in hours. The figure also locates five time periods, which indicates
significant performance variations. From Fig. S6, the CNN model underperformed during weeks
3-9 and 44-51, labeled the "Worst CNN results" in Fig. 2. For weeks 14-22 and 42-44, the CNN
model showed the best forecasting results. Between weeks 29 and 33, the CNN model produced
significant underestimations, labeled "Large under-prediction" in Fig. 2. The figure shows strong
wavelet powers during a 24-hour (daily) period for all variables, the results of strong diurnal
variation of these parameters, which are directly or indirectly controlled by sunlight (e.g.,
temperature, relative humidity, etc.). While the wavelet powers for wind speed were generally
larger than RH, the temperature showed lower, but more consistent daily modes. This finding is
important since the CNN model can more accurately detect specific "patterns" in the temperature
than those in the wind speed and RH. Thus, when the daily modes are stronger in temperature, the
CNN model likely performs better. In contrast, when the daily modes of the meteorological
variables are relatively weak, the CNN model performs poorly (see Fig. 2).
The large coarse modes in the wind speed and RH lead to significant over and
underestimation of the CNN model. Figure S8 shows the polar frequency (influenced by the wind
speed) of the CNN modeling bias in various months. As the figure shows, while southwesterly
winds in August 2017 were associated with relatively large underpredictions boosted by pollution
transport from the Incheon area, north-northwesterly winds with air coming from less urbanized
regions were allied with notable overpredictions.
Figure S9 compares the CNN model predictions with observational data for the seasons
with respect to levels of RH. The figure shows the largest differences in the CNN model
predictions (both over and underpredictions) when the level of RH was close to the extreme (very
high and very low). This finding was particularly evident for the summer months when the model
showed poor performance at capturing high ozone episodes. This finding underscores the
importance of coarse models from the wavelet analysis during the warm months. Directly
indicating the over or underpredictions by the model through these modes, however, is
challenging. For instance, Fig. S10 shows one high ozone episode in July 2017, when the daily
ozone peak exceeded 90ppb on two continuous days at most stations. Here, the overprediction of
the CNN model was associated with high RH while the underprediction was linked to low RH,
indicating more complexity among the relationships between meteorological factors and ozone
formation or depletion.
Another reason for the poor performance of the CNN model during the selected time
period was the relatively large coarse modes (period > 24 hours). The CNN model received
information about only the last day; hence, it was unable to address the bi-daily and weekly trends
with the input data. For instance, for time periods with large underpredictions, coarse modes in the
wind speed were even larger than the daily modes. Thus, employing a longer history would
adequately explain the relationship between wind speed and ozone. In the comparison of the
average wavelet powers in various periods (from daily to weekly modes) of CNN predictions and
observational data, Fig. 3 shows that the powers for both time series match periods of
approximately 24 hours. After 32 hours, however, the wavelet power of the CNN model shrinks
to a relatively constant power while that for the observation reaches local extremums at around 3,
5, and 7 days.
Although wavelet analysis indicates that modes coarser than 24 hours are important
components of the ozone time series, their relationship to CNN model accuracy can be



complicated. Figure 4 compares wavelet powers for both fine and coarse modes with a correlation
coefficient (r) in 25 ozone stations in Seoul. For stations closer to the downtown area (i.e., those
with station numbers under 11), the fine modes had fewer wavelet powers than those for stations
in less urbanized areas, indicating that the relationship between ozone concentrations with local
emissions was evident in the less urbanized areas than it was in the other areas. The coarse modes,
however, varied from station-to-station with relatively higher coarse wavelet power for those in
less urbanized areas. Nonetheless, no evidence points to a clear relationship between either coarse
or fine wavelet modes and the accuracy of the model. Figure 4 shows that the CNN model generally
performed better for stations close to downtown Seoul. Because Seoul has only one meteorological
station, these stations had accessed to more realistic weather parameters in their training/prediction
process.
**3.1.2. Low modeling performance during the nighttime:**
In their discussion of several air quality forecasting models that incorporated machine
learning algorithms, including CNN, deep neural networks, and decision trees, Eslami et al.
(2019a) and Eslami et al. (2019b) claimed that the algorithms encounter a significant modeling
bias while estimating air quality concentrations during the nighttime. This bias reduced the
prediction accuracy of nighttime ozone concentrations, compared to daytime concentrations, by
more than 20%. A similar issue is also encountered by CTMs, even those with complex physical
and chemical equations that explain the diurnal variation of ozone concentrations.
One reason for this modeling bias was likely the result of variation among the
meteorological inputs during the nighttime. Although their absolute values were generally higher
they were during the daytime, the relative frequency of variation was more pronounced during the
nighttime, causing a discontinuity in the learning process of the CNN model. Since both daytime
and nighttime hours were inputs, the CNN model minimized the cost function that contained
"normalized" errors during both daytime and nighttime hours (the cost function was the mean
squared errors or 24-hour ozone predictions at each step). Generally, there are more daytime hours
than nighttime hours (see Fig. S11). Also, the accumulation of $NO_2$ concentrations for these
extreme cases was mainly due to stagnant atmospheric conditions with wind speeds close to their
yearly minimum values (see Fig. S12a for scatter plots with levels of wind speeds). As a result,
the CNN model was vulnerable to characteristic bias in nighttime ozone estimations. As a
customized cost function could be a potential solution to this limitation, it requires further
investigation.
The performance of the CNN model in predicting nighttime ozone concentrations also
suffered because of the misinterpretation of extreme conditions of the input parameters. Figure 5
shows scatter plots that compare CNN predictions and observations by the levels of two important
ozone precursors ($NO_2$ concentrations) and meteorological variables (RH%) separated into
daytime and nighttime. The $NO_2$ concentration was generally higher during the nighttime, when
the ozone concentration was near zero for extreme $NO_2$ values because of conditions amenable to
ozone depletion with the absence of sunlight. Unable to capture this relationship, however, the
CNN model overestimated these cases (See Fig. 5a).
In contrast to the above-mentioned overestimated events, Fig. 5b shows an underestimation
of nighttime ozone when the level of RH% was generally high, primarily during warm days. A
similar pattern occurred when the surface pressure was accounted for (Fig. S12b). Such
underestimated events occurred for two reasons. One is that high (or low) levels of RH% and
surface pressure generally occur at about the same time during the early morning (or late



afternoon), when the planetary boundary layer (PBL) is at its lowest (or highest) level during the
day. In these extreme conditions, the earlier sunrise (or later sunset) during the summer months
established a condition that elevated ozone concentrations. As these events normally occurred only
during short periods of time, the CNN model was not sufficiently trained to capture these
relationships.
**3.2. Case 2: CNN as a post-processing tool in a real-time ozone forecasting system:**
In this case, a generalized bias-correction CNN model introduced by Choi et al. (2019) was
used. Their model is a computationally efficient deep learning-based model that produces more
reliable numerical results. The authors used a deep CNN model to map ozone precursors from
CMAQ and meteorological parameters from the weather research and forecasting (WRF) model
(as input variables) to observe hourly ozone concentrations at a monitoring station (as a target).
Their model, the CMAQ-CNN model, significantly improves the performance of the CMAQ
model in both accuracy and bias. Figures S13 shows the statistical improvements (in correlation,
root mean squared error, and standard deviation) of the CMAQ-CNN model over the CMAQ
model (as a base model) in different months. Figure S14 compares the daily maximum ozone
estimated by CMAQ and CMAQ-CNN in 48 states for which the CMAQ-CNN significantly
moderated the overpredictions of the CMAQ.
It was clear that the likelihood of the CMAQ-CNN model producing accurate results was
strongly associated with the quality of CMAQ forecasts; when CMAQ forecasted hourly ozone
concentrations with a station-specific yearly IOA more than 0.5, the IOA of the CMAQ-CNN
model was more than 0.8 for most cases. The probability of such accuracy was generally unrelated
to that of the CMAQ model. For instance, the CMAQ-CNN model was unable a reach the yearly
IOA=0.8 even though the CMAQ IOA was more than 0.7 (e.g., EPA #101 Tennessee: CMAQ
IOA=0.7; CMAQ-CNN IOA=0.78). In some cases, however, the yearly IOA following the post-
processing approach was less than 0.7 (e.g., EPA #1011 California: CMAQ-CNN IOA=0.63).
Here, we used the distance analysis from DTW to explain (i) why CMAQ-CNN produced
satisfactory results at some stations but not others, and (ii) why it performed poorly at some
stations.
**3.2.1. Satisfactory post-processing scenarios:**
Figure 6 shows the time-series of CMAQ, CMAQ-CNN, and observed daily ozone
concentrations at three EPA stations. These stations were selected because the IOA accuracy of
the CMAQ-CNN model was either more than 0.9 (Fig. 6a and 6b) or 20% more than that of CMAQ
(Fig. 6c). Figure 7 compares the DTW distance analysis of CMAQ and CMAQ-CNN for the same
stations. These are three typical cases of satisfactory improvement by the CMAQ-CNN post-
processing approach:
Figures 6-7(a): Observed ozone concentrations in this California location were higher at the
beginning of the ozone season, followed by relatively steady values ranging
between 20-40ppb. After May, however, CMAQ significantly overestimated daily
ozone concentrations. The overestimation was more pronounced at the end of the
ozone season, resulting in an overall IOA accuracy of 0.73. The DTW distance
analysis showed a consistent distance between CMAQ predictions and observed
values. Because of this consistency, the CMAQ-CNN model recognized the bias
trends in CMAQ, boosting its prediction accuracy by 0.17, even though the large





distance from the CMAQ predictions (mean distance=0.52) mirrored a relatively significant overestimation in the CMAQ-CNN post-processed results.

Figures 6-7(b): Here, the trend in ozone concentrations followed a U-shaped curve in the ozone season because of strong summer winds coming from the large bodies of water near Florida (the North Atlantic Ocean and the Gulf of Mexico). For this station, CMAQ accurately predicted this trend throughout the ozone season with a relatively constant bias from July to September. As a result, the overall accuracy of the IOA was 0.84 for the CMAQ prediction. The CMAQ was also consistent with the DTW analysis, with two distance gaps in July and September (at the beginning and the end of the CMAQ overestimation period). The CMAQ-CNN model, recognizing the adequate performance of the base model in its post-processing algorithm, further improved the IOA accuracy of CMAQ by around 10%.

Figures 6-7(c): The trend of observed ozone showed a steady decrease in this northeastern state because of the significantly cooler summer and fall months. This trend, along with the fewer ozone emission sources surrounding this station resulted in the formation of less ozone during the ozone season. The CMAQ model overestimated ozone concentrations by more than 50% during most of the season with a relatively large mean DTW distance (0.62). The CMAQ-CNN model was able to address this issue because of the consistency of the bias trend in CMAQ predictions (see left panel for DTW distance). Thus, overall, the accuracy of IOA improved by 0.2.

The satisfactory post-processing results using the CMAQ-CNN model were mainly characterized by the regularity of the bias trend in CMAQ as the base model for training the CNN model. As shown by the DTW distance analysis, when the DTW distance of CMAQ predictions from observed values was consistent throughout the ozone season, the CNN model was able to improve the CMAQ results to a reliable level (IOA>0.8). To test this hypothesis, we used the CMAQ-CNN post-processing approach in typical unsatisfactory scenarios.

### 3.2.2. Unsatisfactory post-processing scenarios:

Figure 8 compares the time series of ozone observations with the CMAQ and CMAQ-CNN models at three selected EPA stations. For all of these stations, the CMAQ-CNN model failed to reach a reliable IOA accuracy level of 0.8 while the accuracy of the CMAQ model improved. Figure 9 represents the DTW distance analysis of the two models and the ozone observation for the same stations. Unsatisfactory improvement by the CMAQ-CNN model occurred in the following three cases:

Figures 8-9(a): The ozone trend in this station fluctuated throughout the ozone season with frequent spikes in May, July, and October, primarily the result of biomass burning (Choi et al., 2016). While the CMAQ model predicted ozone concentrations with a relatively small bias (IOA=0.7), the bias trend varied from time to time—that is, trends of under and overpredictions changed frequently. A footprint of these trends, that is, changes in the path of the distance trend, is evident in the DTW analysis. This inconsistency was mirrored in the equivalent DTW analysis for the CMAQ-CNN model by a consistent distance trend, resulting in an unsatisfactory IOA accuracy level (IOA=0.78) with an increased mean DTW distance (0.89 compared to 0.74 for the CMAQ time series).





Figures 8-9(b): The trend in this California location was a relatively constant concentration of ozone generally ranging between 10-30ppb. The CMAQ model significantly overpredicted ozone concentrations throughout the entire time period, mostly the result of the proximity of this station to the Pacific Ocean (San Diego County), which controls the variation in the daily ozone concentration (Pan et al., 2017). The DTW distance analysis shows a significant, yet steady spike in distance between CMAQ and the observation. Thus, even though the CMAQ-CNN significantly improved the accuracy of the CMAQ model (IOA=0.63 compared to CMAQ IOA=0.44), the large distance accounted for the underperformance of the post-processing approach. That also mirrored the consistent distance in the CMAQ-CNN distance trend (see the right panel).

Figures 8-9(c): In this station, the ozone concentration followed an infrequent trend with lows and highs spread indiscriminately across the ozone season, the result of several factors affecting air pollution in this region, including biomass burning, a strong frontal system, and other conditions. As a result, the CMAQ model underperformed with substantial overestimation during most of the time period (IOA=0.55). In addition, the bias of the CMAQ model did not follow as clear a trend as the DTW distance analysis. The CMAQ-CNN model improved the prediction results by more than 10% with a reduced DTW distance (0.27 vs. 0.35 for the CMAQ time series). Nevertheless, the varying ozone trend accompanying the inconsistency in the prediction bias trend resulted in the low overall accuracy of the IOA of the CMAQ-CNN for this station (IOA=0.67).

Unlike the satisfactory cases, the unsatisfactory post-processing results using the CMAQ-CNN model stemmed from the inconsistency in the bias trend found by the DTW distance analysis. Another influential factor was the variability of observed ozone concentrations. Because of the frequent variation in the observational data, it was more complicated to train the CMAQ-CNN model so that it addressed the bias in the CMAQ model. The geographical location of a station was also an important factor in the improvement level of the post-processing approach. Proximity to the large body of water and/or sources from biomass burning during the ozone season were among the influential geographical features. Also, as Figs. 8-9 show, the DTW distances of the CMAQ-CNN predictions from the observed ones followed a consistent trend. Therefore, the information in Figs. 6-7 indicate that a secondary post-processing model might be a possible solution to boosting prediction accuracy.

## 4. Conclusion:

Various applications of deep learning algorithms, particularly convolutional neural networks, have universally been applied in the field of atmospheric sciences, especially in air quality forecasting systems. Although such applications supported easy-to-use, computationally-efficient frameworks and flexible capabilities appeared to generate accurate prediction results, the risk of exaggerated expectations may be a cause for concern. In an effort to elucidate both the advantages and limitations of deep learning models in air quality forecasting (AQF) systems, this paper addressed several common issues raised by the use of these models.

To explore the limitation, we chose two applications of two similar CNN models. (i) CNN as an independent real-time AQF; and (ii) CNN as a post-processing model of a state-of-the-art dynamical model, the Community Multi-scale Air Quality Model (CMAQ). For both cases, the CNN model resulted in an acceptable 24-hour in advance, hourly ozone concentration prediction





with an index of agreement (IOA) of more than 0.8 for two networks of monitoring stations in South Korea and the United States. We selected two powerful statistical data analytic techniques—wavelet transform and dynamic time warping (DTW)—to identify the limitations of the proposed models in both cases. By applying these techniques, researchers find discrepancies in the input data and their temporal trends and thus gain awareness of the limitations of deep learning models.

When the CNN model was used as a real-time AQF system in South Korea, it underperformed during both cold months and high ozone episodes. In these scenarios, we found that the fine wavelet modes (daily and hourly) were relatively weaker than they were in other conditions. Also, when the coarse modes were strong, the predictions of the CNN model were fraught with a large number of errors. We also found that the model underperformed during the nighttime hours, the results of an undertrained model and extreme values of the input parameters during the nighttime.

For the post-processing CNN model, the level of improvement depended on the DTW distance of the CMAQ model to the observations. When the calculated distance followed a consistent trend, the post-processing model was able to address the bias of CMAQ, independent from its accuracy level or error range. When such consistency was absent or when observed ozone varied frequently, however, the errors in the CMAQ model were mirrored in the results of the post-processing model.

Given this discussion of the limitations of deep learning models, we suggest that researchers configure their deep learning models based on temporal trends within the input parameters, geographical locations, and variation frequency of target pollutants. To predict ambient hourly ozone concentrations, we have restricted our discussions to a multi-output regression problem in supervised settings. While our study approach might be valid for other supervised algorithms, we leave a detailed study of other supervised methods for future work.

**Code availability.** The code for the algorithm development, evaluation, and statistical analysis is freely available for non-commercial research purposes by contacting the corresponding author.

**Supplement.** The supplementary document related to this article is available.

**Author contribution.** E.E., Y.C, Y.L., A.S., and A.K.S. contributed to the design and implementation of the research, to the analysis of the results. E.E. took the lead in writing the manuscript with inputs from Y.C, Y.L., A.S., and A.K.S.. Y.C. supervised the project. E.E. and A.S. prepared the modeling input data and optimized the python codes. All authors discussed the results and commented on the manuscript and contributed to the final version of the manuscript.

**Competing interest.** The authors declare no competing financial and/or non-financial interests in relation to the work described.

**Acknowledgment**
This study was supported by the High Priority Area Research Seed Grant of the University of Houston. The authors also express their gratefulness to Drs. Wonbae Jeon and Shuai Pan whose prepared the 4-year CMAQ and SMOKE runs for the TCEQ Project No. 582-15-54181-09, which were used in this study.



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





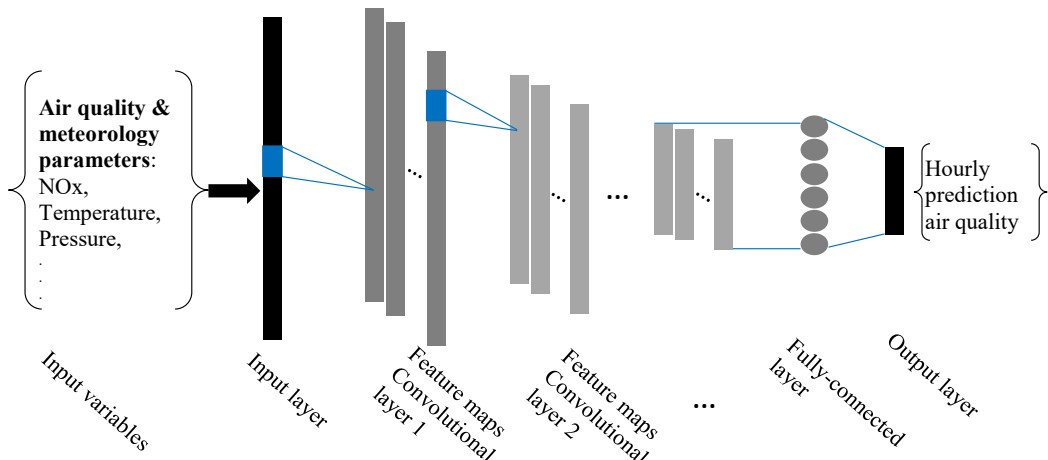

**Figure 1.** Schematic of the deep CNN model in our approach.

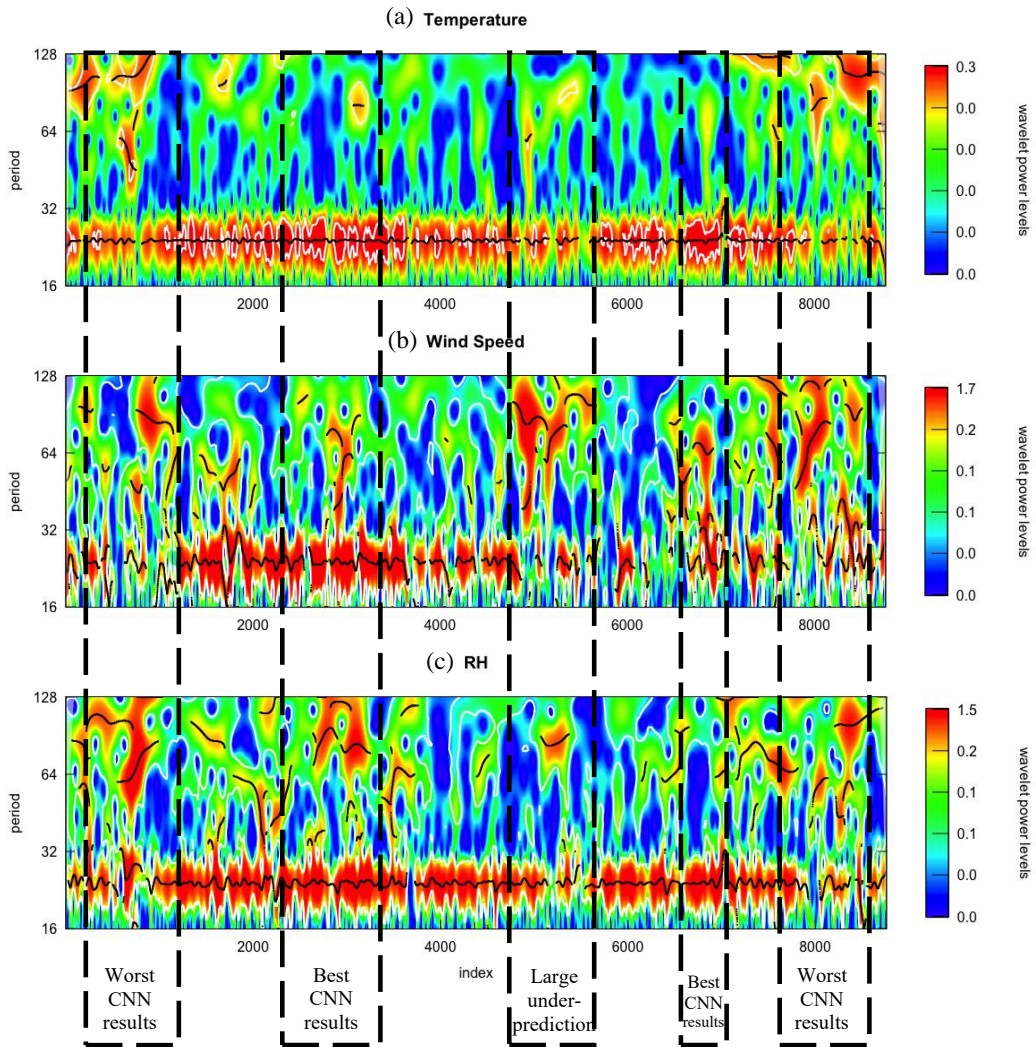

**Figure 2.** Wavelet power transform of (a) temperature, (b) wind speed, and (c) RH% for 2017 in Seoul, South Korea.

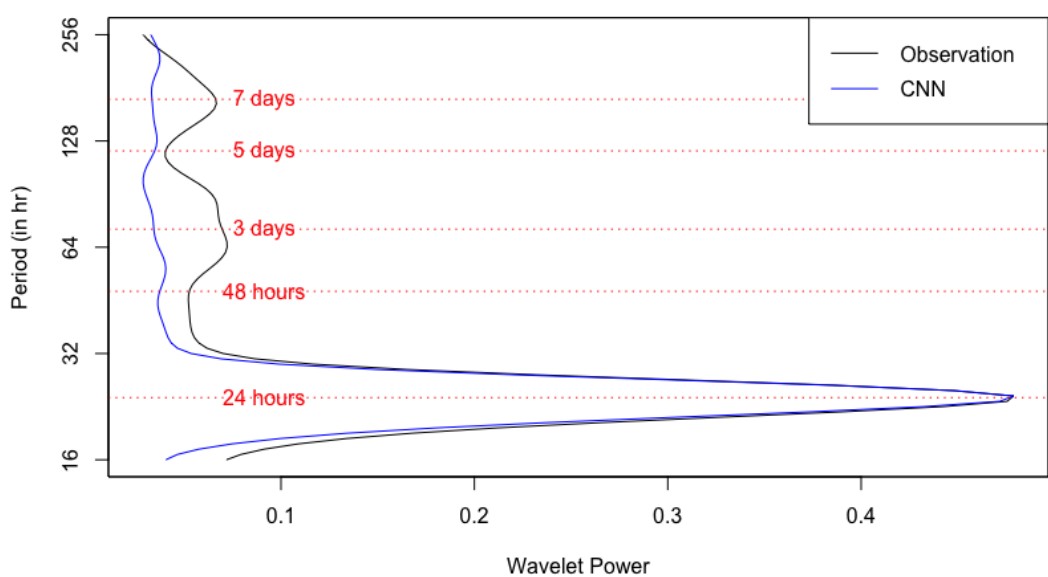

**Figure 3.** Wavelet power for various time periods (modes) for CNN predictions and observations.

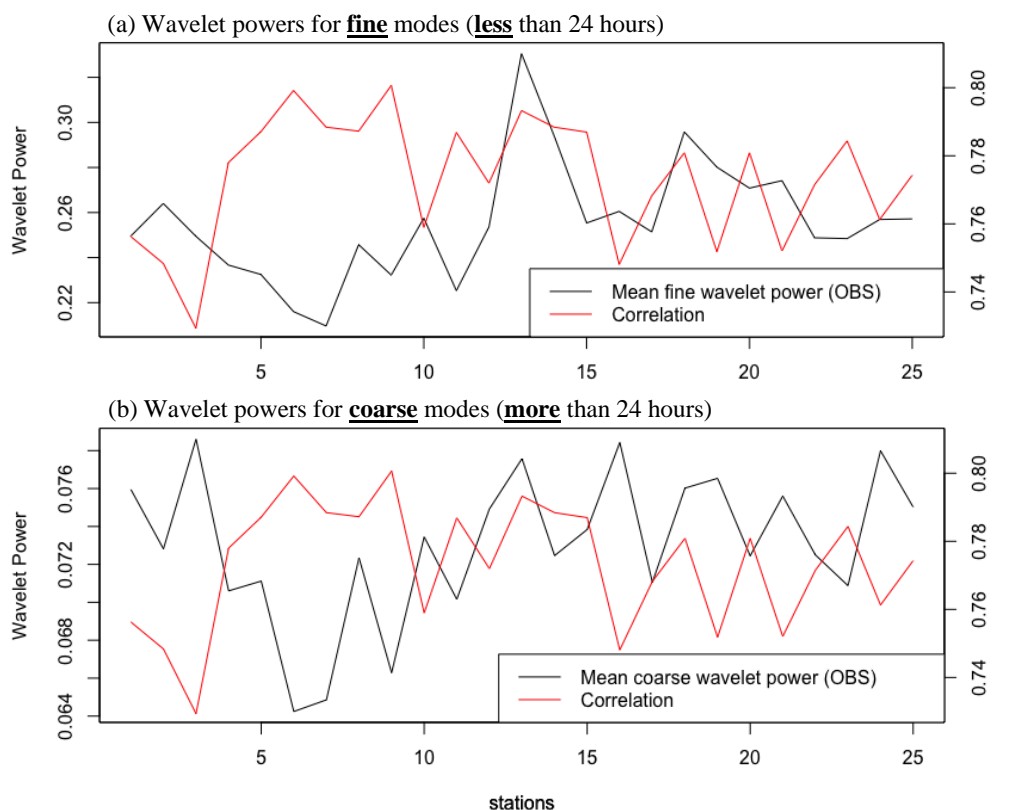

**Figure 4.** Relationship between (a) fine and (b) coarse wavelet power modes and correlation coefficients in all stations in Seoul, South Korea.

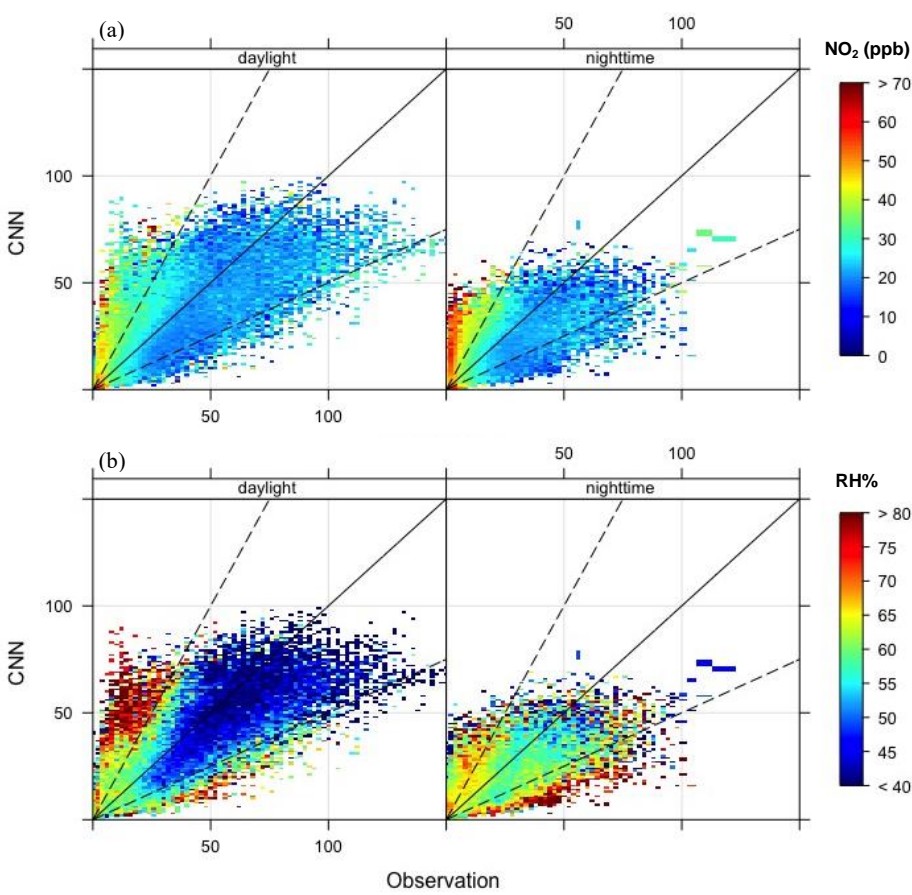

**Figure 5.** Scatter plots comparing CNN predictions and observations with respect to levels of (a) NO₂ concentrations and (b) RH%.



**Figure 6.** Comparison of the time series of CMAQ and CMAQ-CNN predictions for EPA stations (a) #3001 (California), (b) #33 (Florida), and (c) #4 (Vermont).



**Figure 7.** Comparison of the distance analysis of CMAQ and CMAQ-CNN predictions for EPA stations (a) #3001 (California), (b) #33 (Florida), and (c) #4 (Vermont).

**Figure 8.** Comparison of the distance analysis of CMAQ and CMAQ-CNN predictions for EPA stations (a) #101 (Tennessee), (b) #1011 (California), and (c) #9008 (Oklahoma).





**Figure 9.** Comparison of the distance analysis of CMAQ and CMAQ-CNN predictions for EPA stations (a) #101 (Tennessee), (b) #1011 (California), and (c) #9008 (Oklahoma).