# Peer review of "Using wavelet transform and dynamic time warping to identify the limitations of the CNN model as an air quality forecasting system"

_Geoscientific Model Development, 2019_

## Referee Comment (RC1) · Anonymous Referee #1 · 25 Mar 2020

The paper provides two case studies critiquing CNN models trained for AQF applications. The first ML model is directly an estimator, the second is used as a corrector for a CMAQ model. The authors use wavelet modal decomposition and a shape-invariant distance metric as analysis tools to find discrepancies in the model predictions and trace them back to environmental factors. This analysis is valuable and interesting in itself. Both positive and negative results are provided.

I encourage the authors to rethink the vision of this paper. What is the central thesis of the paper? Does CNNs work better as post-processing tools rather than raw predictors? Are model biases inevitable in these applications no matter the configuration?

The authors diagnose important limitations of CNN models trained on their data but very few thoughts are offered for interested researches as to how to fix these issues. (except maybe in the conclusions). For example if there is a significant difference in the accuracy of the model in nighttime vs daytime, how does a single model compare to two models trained separately on subsets of data (day/night). If your analysis shows hidden correlations between the error and RH% how can you incorporate that into the input data?

When the model is not performing well, insufficient training (as suspected by authors) is only one possible cause. Another possibility may be under parametrization, such that the model is not complex enough to capture the details of special cases. I think providing error measures on the training data and comparing them with test data can illuminate the source of underperformance.

The authors state on line 46 : "Inevitably, a consequence of such enthusiasm in the field is the risk of exaggerated expectations, fueled by results focusing on the general performance of ML models compared to that of conventional statistical models" and give their previous works as examples. At the very least this assertion needs a more detailed explanation.

---

## Referee Comment (RC2) · Anonymous Referee #2 · 12 Apr 2020

This paper proposed a wavelet-based approach to evaluate the advantage and disadvantages of a typical deep learning model, convolutional neural network, in air quality forecasting (AQF). They used wavelet transform to identify the causes of the poor performances of CNN and find that when fine wavelet modes are relatively weak or coarse wavelet modes are strong, CNN forecasts will be less accurate. This finding is very important for the community to understand the drawbacks of deep learning and be aware of them when using it together with conventional numeric air quality models. The paper has a clear design and the proposed idea and the subsequent experiments are presented very well.

---

## Author Comment (AC1) · 3 Jun 2020

**Responses to the comments of Referee #1:**

We would like to thank the reviewer for his/her time and effort for reviewing this manuscript. Please find below our responses.

*Referee #1:*

*The paper provides two case studies critiquing CNN models trained for AQF applications. The first ML model is directly an estimator, the second is used as a corrector for a CMAQ model. The authors use wavelet modal decomposition and a shape-invariant distance metric as analysis tools to find discrepancies in the model predictions and trace them back to environmental factors. This analysis is valuable and interesting in itself. Both positive and negative results are provided.*

*I encourage the authors to rethink the vision of this paper. What is the central thesis of the paper? Does CNNs work better as post-processing tools rather than raw predictors? Are model biases inevitable in these applications no matter the configuration?*

**Response:**

To respond to your suggestion and comments, the following statements are offered:

- Despite the enormous success of the convolutional neural network (CNN) algorithm in numerous applications, certain issues related to its applications in air quality forecasting (AQF) require further analysis and discussion. Our main goal in this paper was to discuss some of these issues is a few practical applications. In order to discuss these issues analytically, we used wavelet transform and dynamic time warping (DTW), as powerful mathematical tools for time-series analysis and models. Based on the findings that were presented in the paper, these tools are extremely helpful not only in understanding the issues with machine learning models but also in fine-tuning them to improve their performances with a scientific point of view. Awareness of the limitations in CNN models will enable scientists to develop more accurate regional or local air quality forecasting systems by identifying the affecting factors in high concentration episodes.
- We discuss the general issues of the CNN model in two common applications: (i) a real-time AQF model, and (ii) a post-processing tool in a dynamical AQF model, the Community Multi-scale Air Quality Model (CMAQ). As the referee correctly stated, these examples are fundamentally different in terms of execution, one being raw predictor (statistical approach) while other being a post-processor (hybrid approach). Since both models are commonly being used as a real-time air quality prediction systems, we discussed their issues individually to broaden researchers' view on certain issues that one may encounter in executing either of them. Thus, it will prove both machine learning researchers and atmospheric scientists with multiple candidate models and analytical tools to develop any specific model of their choice.

*The authors diagnose important limitations of CNN models trained on their data but very few thoughts are offered for interested researches as to how to fix these issues. (except maybe in the conclusions). For example if there is a significant difference in the accuracy of the model in nighttime vs daytime, how does a single model compare to two models trained separately on subsets of data (day/night). If your analysis shows hidden correlations between the error and RH% how can you incorporate that into the input data?*

*When the model is not performing well, insufficient training (as suspected by authors) is only one possible cause. Another possibility may be under parametrization, such that the model is not complex enough to capture the details of special cases. I think providing error measures on the training data and comparing them with test data can illuminate the source of underperformance.*

**Response:**

To respond to your suggestion and comments, following explanations are stated:

- Based on our findings in the base studies presenting the aforementioned CNN models, in both cases, the CNN model shows promising accuracy for ozone prediction, 24 hours in advance, in both the United States and South Korea. However, similar to other data-driven prediction tools, in a CNN model, the out-of-sample prediction error is almost always greater than the in-sample prediction error. Thus, since both CNN models were designed as a real-time air quality prediction models, the prediction error is inevitable, even though (i) both models were configured for optimum performance (based on the input or training samples), and (ii) in development of both models, careful cross-validation processes were followed to mitigate any systematic biases. In addition, a comprehensive explanation can be found in our previous works, including but not limited to, the potential reasons for underperformance of the CNN model, modeling configuration and fine-tuning processes, training and validation process, arrangements of input variables, scenarios to improve the modeling accuracy, etc. Please refer to Eslami et al. (2019a, 2019b, 2019c), Choi et al. (2019), Sayeed et al. (2020), and Lops et al. (2019), and the discussion within. The authors will be delighted to provide additional explanations if nessaccry to accommodate the referee's comments and suggestions.

- For one case (raw prediction model), we use the wavelet transform to determine the reasons behind the poor performance of CNN during the nighttime, cold months, and high ozone episodes. We find that when fine wavelet modes (hourly and daily) are relatively weak or when coarse wavelet modes (weekly) are strong, the CNN model produces less accurate forecasts. For the other case (post-processing model), we use the DTW distance analysis to compare post-processed results with their CMAQ counterparts (as a base model). For CMAQ results that show a consistent DTW distance from the observation, the post-processing approach properly addresses the modeling bias with predicted IOAs exceeding 0.85. When the DTW distance of CMAQ-vs-observation is irregular, the post-processing approach is unlikely to perform satisfactorily. We are currently working on an individual study to use the findings of this study to fine-tune both CNN models. In our work-in-progress study, we will provide a

78    practical approach in infusing scientific angel of air quality time-series into CNN prediction
79    models using advanced analytical tools.

80
81    *The authors state on line 46 : "Inevitably, a consequence of such enthusiasm in the field is the risk*
82    *of exaggerated expectations, fueled by results focusing on the general performance of ML models*
83    *compared to that of conventional statistical models" and give their previous works as examples.*
84    *At the very least this assertion needs a more detailed explanation.*
85
86    • To address the referee's comment on line 46, we changed the sentence as highlighted in the
87        following, and we applied the changes in the manuscript:
88        o  However, the focus of these studies was the general performance of the model ML models
89            compared to that of conventional statistical models rather than identifying the shortcoming
90            of such models in explaining the uncertainties of prediction models. Such examples can be
91            found in studies by Eslami et al. (2019a, 2019b, 2019c), Choi et al. (2019), Sayeed et al.
92            (2020), and Lops et al. (2019). To achieve more reasonable outcomes, we must first explore
93            the current challenges we face when forecasting ambient air quality and then assess how or
94            even whether ML models can address the challenges to produce more accurate forecasting.

---

## Author Comment (AC2) · 3 Jun 2020

**Responses to the comments of Referee #2:**

*Referee #2:*

*This paper proposed a wavelet-based approach to evaluate the advantage and disadvantages of a typical deep learning model, convolutional neural network, in air quality forecasting (AQF). They used wavelet transform to identify the causes of the poor performances of CNN and find that when fine wavelet modes are relatively weak or coarse wavelet modes are strong, CNN forecasts will be less accurate. This finding is very important for the community to understand the drawbacks of deep learning and be aware of them when using it together with conventional numeric air quality models. The paper has a clear design and the proposed idea and the subsequent experiments are presented very well.*

**Response:**

We would like to thank the reviewer for his/her time and effort for reviewing this manuscript.

---

## Referee Report (RR1)

**Review of 'Using wavelet transform and dynamic time warping to identify the limitations of the CNN model as an air quality forecasting system' by Eslami et al.**

In this paper, Eslami et al. present a method based on wavelet transform and dynamic time warping (DTW) to characterize the quality of a machine-learning (ML) algorithm (convolutional neural network, CNN) for air quality forecasting (AQF). Using the example of two AQF applications, they show how wavelet transform and DTW can provide new insights into the strengths and weaknesses of the CNN model.

Better understanding the potential and limitations of ML algorithms for AQF applications is a topic that is rapidly gaining importance given the explosion of ML applications in this area. This paper makes a valuable contribution to this discussion by presenting a powerful analytical tool that can effectively highlight conditions under which the employed ML algorithm fails to produce satisfactory results. As such, the manuscript is highly suitable for publication in GMD. However, in its current form there are still some issues regarding the main message of the paper and how wavelet transform and DTW can be used to improve error characterization of ML applications.

For instance, the authors simultaneously say that the tested CNN models have 'significant limitations' and 'show promising accuracy', and generally seem to switch between the view that the ML model is either 'bad' or 'good'. In reality, the CNN models – like chemical transport models – perform very well under some conditions and poorly under others. One of the powerful elements of the discussed statistical analysis tools is that they offer a method to identify these conditions and thus help the model developers better understand the strengths and limitations of the ML algorithms. This information also helps identify how the ML model might be improved, which is very powerful. The authors should stress this more clearly.

Another point that needs more discussion is the time dimension. The used CNN models seem to use snapshots of time-series data as inputs (rather than a window of the time-series) and are thus not designed to learn temporal relationships. This should be stated more clearly, as it means that the wavelet transform and DTW offer an assessment of a feature that is not directly optimized by the ML algorithm (which is a good thing).

Minor comments:
- Page 4, line 100: 'general inability of the machine learning model' seems a bit too harsh. I suggest to rephrase this.
- Page 5, line 124. Should be Figure 1, not Figure 3.
- Page 6, line 201: Please provide the definition of index of agreement
- Page 6, line 213: I'd be careful with the statement that NOx and VOC emissions are constant in time. These emissions have large diurnal and seasonal cycles.
- Page 7, line 251ff: maybe worth mentioning here the potential of long short-term memory (LSTM) algorithms to incorporate time dependency in the training?

---

## Author Response (AR2)

**Response to Reviewer:**
**Title: Using wavelet transform and dynamic time warping to identify the limitations of**
**the CNN model as an air quality forecasting system**
**Author(s): Ebrahim Eslami et al.**
**MS No.: gmd-2019-346**
**MS Type: Model evaluation paper**
**Iteration: Minor Revision**
**Responses to the comments of Referee:**
We would like to thank the reviewer for his/her time and effort in reviewing this manuscript. Please
find below our responses.
*Referee:*
*In this paper, Eslami et al. present a method based on wavelet transform and dynamic time*
*warping (DTW) to characterize the quality of a machine-learning (ML) algorithm (convolutional*
*neural network, CNN) for air quality forecasting (AQF). Using the example of two AQF*
*applications, they show how wavelet transform and DTW can provide new insights into the*
*strengths and weaknesses of the CNN model.*
*Better understanding the potential and limitations of ML algorithms for AQF applications is a*
*topic that is rapidly gaining importance given the explosion of ML applications in this area. This*
*paper makes a valuable contribution to this discussion by presenting a powerful analytical tool*
*that can effectively highlight conditions under which the employed ML algorithm fails to produce*
*satisfactory results. As such, the manuscript is highly suitable for publication in GMD. However,*
*in its current form there are still some issues regarding the main message of the paper and how*
*wavelet transform and DTW can be used to improve error characterization of ML applications.*
*For instance, the authors simultaneously say that the tested CNN models have 'significant*
*limitations' and 'show promising accuracy', and generally seem to switch between the view that*
*the ML model is either 'bad' or 'good'. In reality, the CNN models – like chemical transport*
*models – perform very well under some conditions and poorly under others. One of the powerful*
*elements of the discussed statistical analysis tools is that they offer a method to identify these*
*conditions and thus help the model developers better understand the strengths and limitations of*
*the ML algorithms. This information also helps identify how the ML model might be improved,*
*which is very powerful. The authors should stress this more clearly.*
*Another point that needs more discussion is the time dimension. The used CNN models seem to*
*use snapshots of time-series data as inputs (rather than a window of the time-series) and are thus*
*not designed to learn temporal relationships. This should be stated more clearly, as it means that*
*the wavelet transform and DTW offer an assessment of a feature that is not directly optimized by*
*the ML algorithm (which is a good thing).*
**Response:**
To respond to your suggestion and comments, the following section was added to the revised
manuscript:
**3.3. Discussion:**
Despite the enormous success of the convolutional neural network (CNN) algorithm in
numerous applications, certain issues related to its applications in air quality forecasting (AQF)

require further analysis and discussion. Our main goal in this paper was to discuss some of these issues is a few practical applications. To discuss these issues analytically, we used wavelet transform and dynamic time warping (DTW) as powerful mathematical tools for time-series analysis and models. Based on the findings that were presented in the paper, these tools are beneficial not only in understanding the issues with machine learning models but also in fine-tuning them to improve their performances with a scientific point of view. Awareness of the limitations in CNN models will enable scientists to develop more accurate regional or local air quality forecasting systems by identifying the affecting factors in high concentration episodes.

Based on our findings in the base studies presenting the aforementioned CNN models, in both cases, the CNN model shows reasonable accuracy for ozone prediction, 24 hours in advance, in two geographical locations (the United States and South Korea). However, similar to other data-driven prediction tools, in a CNN model, the out-of-sample prediction error is almost always greater than the in-sample prediction error. Thus, since both CNN models were designed as a real-time air quality prediction models, the prediction error is inevitable, even though (i) both models were configured for optimum performance (based on the input or training samples), and (ii) in development of both models, cross-validation processes were followed to mitigate any systematic biases. However, the underperformance of the CNN model was dependent on several factors, including modeling configuration (e.g., the depth of CNN model), arrangements of input variables (e.g., number of previous days as inputs), the day of the week (e.g., weekdays versus weekdays), the hour of the day (e.g., daytime versus nighttime) (see Eslami et al. (2019a, 2019b, 2019c), Choi et al. (2019), Sayeed et al. (2020), and Lops et al. (2019), and the discussion within).

Here, we discussed the general limitations of the CNN model in two common applications: (i) a real-time AQF model, and (ii) a post-processing tool in a dynamical AQF model (i.e., CMAQ). These examples are fundamentally different in terms of execution, one being a raw predictor (statistical approach) while the other being a post-processor (hybrid approach). Since both models are commonly being used as a real-time air quality prediction system, we discussed their issues individually to explain certain issues that one may encounter in executing either of them. Thus, it will provide both machine learning researchers and atmospheric scientists with multiple candidate models and analytical tools to develop any specific model of their choice.

For one case (raw prediction model), we used the wavelet transform to determine the reasons behind the poor performance of CNN during the nighttime, cold months, and high ozone episodes. We find that when fine wavelet modes (hourly and daily) were relatively weak or when coarse wavelet modes (weekly) were strong, the CNN model produced less accurate forecasts. Since the CNN model has used only one precious day of air quality and meteorological parameters, neither the coarse patterns (e.g., weekly) were used as a prediction feature, nor any connection between different time-series windows (as is revealed in a wavelet transform analysis) was considered. Thus, the wavelet transform can be helpful as a complementary tool in filling these gaps in a CNN prediction model development. It should be noted that long short-term memory (LSTM) model can potentially incorporate some of the aforementioned time-dependencies (e.g., bi-daily or weekly). However, the focus of this study is to address such a limitation in a CNN model as the choice of the ML model.

For the other case (post-processing model), we used the DTW distance analysis to compare post-processed results with their CMAQ counterparts (as a base model). For those CMAQ results with a consistent DTW distance from the observation, the post-processing approach properly addressed the CMAQ modeling bias with predicted IOAs exceeding 0.85. When the DTW distance of CMAQ-vs-observation is irregular, the post-processing approach is unlikely to perform satisfactorily. Even though the CMAQ-CNN model has included several chemical components
and meteorological variables as its inputs, there was no input feature representing CMAQ's own
accuracy. By comparing a history of CMAQ results in different geographical locations with
available observation data, the DTW can provide an 'irregularity' index as an additional input
feature.
**Response:**
To respond to your suggestion and comments, the following modifications were made in the
manuscript:
*Referee:*
*Minor comments:*
*- Page 4, line 100: 'general inability of the machine learning model' seems a bit too harsh.*
*I suggest to rephrase this.*
Response: "general inability" has been changed to "certain limitations."
*- Page 5, line 124. Should be Figure 1, not Figure 3.*
Response: Thanks. The figure citation in the text has been changed.
*- Page 6, line 201: Please provide the definition of index of agreement*
Response: The following statement has been added to the manuscript.
Note that IOA is a standardized measure of the degree of model prediction error and varies between
0 and 1. The agreement value of 1 indicates a perfect match, and 0 indicates no agreement at all.
*- Page 6, line 213: I'd be careful with the statement that NOx and VOC emissions are*
*constant in time. These emissions have large diurnal and seasonal cycles.*
Response: Thanks for a good point. The following modification has been made in the
manuscript.
Compared with meteorological variables, emission sources from volatile organic compounds
(VOCs) and NOx are experiencing less variability in time. Thus, meteorological variables play an
important role in governing the variation of the ozone at different times throughout the year
*- Page 7, line 251ff: maybe worth mentioning here the potential of long short-term*
*memory (LSTM) algorithms to incorporate time dependency in the training?*
Response: The following statement has been added to the manuscript in 4th paragraph in the newly
added Discussion section (Section 3.3, lines 476-479).
It should be noted that long short-term memory (LSTM) model can potentially incorporate some
of the aforementioned time-dependencies (e.g., bi-daily or weekly). However, the focus of this

[revised manuscript text omitted]
. Our main goal in this paper was to discuss some of these issues is a few practical applications. To discuss these issues analytically, we used wavelet transform and dynamic time warping (DTW) as powerful mathematical tools for time-series analysis and models. Based on the findings that were presented in the paper, these tools are extremely helpful not only in understanding the issues with machine learning models but also in fine-tuning them to improve their performances with a scientific point of view. Awareness of the limitations in CNN models will enable scientists to develop more accurate regional or local air quality forecasting systems by identifying the affecting factors in high concentration episodes.

Based on our findings in the base studies presenting the aforementioned CNN models, in both cases, the CNN model shows reasonable accuracy for ozone prediction, 24 hours in advance, in two geographical locations (the United States and South Korea). However, similar to other data-driven prediction tools, in a CNN model, the out-of-sample prediction error is almost always greater than the in-sample prediction error. Thus, since both CNN models were designed as a real-time air quality prediction models, the prediction error is inevitable, even though (i) both models were configured for optimum performance (based on the input or training samples), and (ii) in development of both models, cross-validation processes were followed to mitigate any systematic biases. However, the underperformance of the CNN model was dependent on several factors, including modeling configuration (e.g., the depth of CNN model), arrangements of input variables (e.g., number of previous days as inputs), the day of the week (e.g., weekdays versus weekdays), the hour of the day (e.g., daytime versus nighttime) (see Eslami et al. (2019a, 2019b, 2019c), Choi et al. (2019), Sayeed et al. (2020), and Lops et al. (2019), and the discussion within).

Here, we discussed the general limitations of the CNN model in two common applications: (i) a real-time AQF model, and (ii) a post-processing tool in a dynamical AQF model (i.e., CMAQ). These examples are fundamentally different in terms of execution, one being a raw predictor (statistical approach)

while the other being a post-processor (hybrid approach). Since both models are commonly used as a real-
time air quality prediction system, we discussed their issues individually to explain specific issues that one
may encounter in executing either of them. Thus, it will provide both machine learning researchers and
atmospheric scientists with multiple candidate models and analytical tools to develop any specific model
of their choice.

For one case (raw prediction model), we used the wavelet transform to determine the reasons behind
the poor performance of CNN during the nighttime, cold months, and high ozone episodes. We find that
when fine wavelet modes (hourly and daily) were relatively weak or when coarse wavelet modes (weekly)
were strong, the CNN model produced less accurate forecasts. Since the CNN model has used only one
precious day of air quality and meteorological parameters, neither the coarse patterns (e.g., weekly) were
used as a prediction feature, nor any connection between different time-series windows (as is revealed in a
wavelet transform analysis) was considered. Thus, the wavelet transform can be helpful as a complementary
tool in filling these gaps in a CNN prediction model development. It should be noted that long short-term
memory (LSTM) model can potentially incorporate some of the aforementioned time-dependencies (e.g.,
bi-daily or weekly). However, the focus of this study is to address such a limitation in a CNN model as a
choice of the ML model.

For the other case (post-processing model), we used the DTW distance analysis to compare post-
processed results with their CMAQ counterparts (as a base model). For those CMAQ results with a
consistent DTW distance from the observation, the post-processing approach properly addressed the
CMAQ modeling bias with predicted IOAs exceeding 0.85. When the DTW distance of CMAQ-vs-
observation is irregular, the post-processing approach is unlikely to perform satisfactorily. Even though the
CMAQ-CNN model has included several chemical components and meteorological variables as its inputs,
there was no input feature representing CMAQ's own accuracy. By comparing a history of CMAQ results
in different geographical locations with available observation data, the DTW can provide an 'irregularity'
index as an additional input feature.

[revised manuscript text omitted]